# Efficacy of a respiratory syncytial virus vaccine candidate in a maternal immunization model

Jorge C.G. Blanco[1], Lioubov M. Pletneva[1], Lori McGinnes-Cullen[2], Raymonde O. Otoa[1], Mira C. Patel[1], Lurds R. Fernando[1], Marina S. Boukhvalova[1] & Trudy G. Morrison[2]

Respiratory syncytial virus (RSV) is the most common cause of bronchiolitis in infants. Maternal immunization is an option to increase maternal antibody levels and protect infants from infection. Here we assess the efficacy of virus-like particle (VLP) vaccine candidates containing stabilized pre-fusion (pre-F) or post-fusion (post-F) conformations of the RSV F protein and the attachment RSV G protein in a maternal immunization model using cotton rats. VLP vaccines containing RSV F and G proteins strongly boost pre-existing RSV immunity in dams preventing their perinatal drop in immunity. Boosting is stronger for the pre-F VLP than for the post-F VLP or purified subunit F protein vaccines, giving an advantage on mothers' protection. VLP immunization of dams provides significant protection to pups from RSV challenge and reduced pulmonary inflammation. Collectively, our results show that a VLP vaccine with RSV F and G proteins is safe and effective for maternal and adult vaccination.

[1] Sigmovir Biosystems Inc., 9610 Medical Center Drive, Suite 100, Rockville, MD 20850, USA. [2] Department of Microbiology and Physiological Systems, Program of Immunology and Microbiology, University of Massachusetts Medical School, Worcester, MA 01655, USA. Correspondence and requests for materials should be addressed to J.C.G.B. (email: j.blanco@sigmovir.com)

Maternal vaccination against infections that affect mothers and infants is now routine and includes vaccines against influenza, tetanus, and pertussis. However, a maternal vaccine for respiratory syncytial virus (RSV) is not available in spite of a significant need. RSV is the single most important cause of acute viral respiratory disease in infants and young children frequently resulting in hospitalization of infants in the US and in significant mortality rates in developing countries[1]. Global RSV-associated acute lower respiratory infections in children under age 5 accounts for ~33 million cases annually with ~10% of them resulting in hospitalization and an in-hospital mortality rate of ~1–3%[2].

Maternal transfer of RSV immunity to infants is likely protective during the first months of life since the level of RSV immunity in mothers during the last trimester of pregnancy and the time of delivery correlates with levels of protection of infants[3]. Thus, increasing maternal RSV immunity during pregnancy could potentially extend the time of protection of their infants. Moreover, recent evidence from human studies and experimentally corroborated by us[4–6] suggested that the period just prior to delivery may be accompanied by a transient drop in immunity in pregnant females. This finding suggests a window of vulnerability in pregnant mothers that may need to be considered during the rational design of RSV maternal vaccines. Thus, a maternal RSV vaccine that could benefit both mothers and infants would be desirable.

A complication for development of a maternal RSV vaccine is that the vast majority of humans has experienced RSV infection by 2–5 years of age[7]. While naturally occurring pre-existing immunity is incompletely protective, it could well impact the effectiveness of a vaccine. Thus, a successful vaccine candidate must stimulate high titers of neutralizing antibody (NA) in the face of pre-existing immunity. Model animal systems using naive animals are instrumental in initial immunogenicity, safety, and efficacy vaccine studies, but may be suboptimal for directly mimicking human responses to RSV vaccines, which will virtually always be in the context of previous exposure(s).

Use of live attenuated virus vaccines, which might overcome pre-existing immunity, are not recommended by regulatory entities to be used during pregnancy due to safety concerns. Virus-like particles (VLPs) are increasingly recognized as a safe and effective platform for developing vaccines against viral diseases[8]. VLPs are virus-sized particles composed of repeating structures on their surfaces and in their cores, structures that mimic those of infectious viruses, which contribute to VLPs' immunogenicity. VLPs are formed by assembly of viral structural proteins and sometimes lipids into particles without the incorporation of the viral genome. Thus, VLPs are incapable of replication. The surface glycoproteins of enveloped viruses are folded and inserted into VLP membranes typical of an infectious virus, thereby retaining antigenic epitope conformation.

We have recently described a Newcastle disease virus (NDV)-based VLP vaccine against RSV that stimulates safe and protective immune responses in mice and in cotton rats[9,10]. In addition, we have also demonstrated that cotton rats are a good model for testing the efficacy and safety of maternal vaccination against RSV[6,11].

Here we show the effect of previous RSV infection on induction of anti-RSV immunity by immunization with VLPs containing different conformational forms of the RSV F protein and the G protein in cotton rat females during gestation, and the ability of these vaccines to provide protection from RSV challenge of offspring. VLP immunization significantly increases NA titers in both RSV-seropositive pregnant females and their litters. Importantly, litters of these animals are notably better protected from RSV replication in the lungs than offspring of RSV naive,

vaccinated animals. Furthermore, the transient immunosuppression in pregnant females shortly before delivery is prevented by VLP immunization. Pulmonary protection and serology in pups is more robust in animals vaccinated with VLPs than animals vaccinated with purified F proteins. Pre-F protein-containing VLPs provide an immunologic advantage to dams and afford a moderate improvement in nasal protection of pups compared to post-F-containing VLPs or to vaccine preparations comprised of purified subunit F protein.

## Results

**Properties of VLPs used for immunization.** VLPs used for immunization are based on NDV core nucleocapsid (NP) and matrix (M) proteins and contain the ectodomain of the RSV G protein and the stabilized pre-fusion or post-fusion forms of the RSV F protein. These VLPs were produced in avian ELL-0 cells as previously described[12] to generate stocks of VLP-H/G + Pre-F/F or VLP-H/G + Post-F/F (abbreviated as Pre-F VLP and Post-F VLP, respectively)[9,13]. The RSV F and G protein content and RSV F protein conformation in the two purified VLP preparations were quantified and characterized by western analysis and antibody binding to the purified VLPs (Supplementary Figs. 1–3)[9,13]. In addition, the soluble post-F and pre-F proteins used for vaccination in this study were characterized by binding to form-specific monoclonal antibodies and quantified by western analysis (Supplementary Fig. 4).

**Infection and immunization.** To study vaccine efficacy in naive and pre-exposed animals, one set of female cotton rats was primed by intranasal (i.n.) RSV infection (RSV primed), while a second set of animals was not RSV primed (naive) (Fig. 1 and Supplementary Table 1). All animals were set up for breeding and vaccinated during pregnancy with VLPs containing G and pre-F or post-F proteins intramuscularly (i.m.). Additionally, vaccinations with purified F proteins in pre-F or post-F conformation were also included to immunize pregnant females for comparison. All vaccinations were performed once using equal amount of F protein in the preparations (10 μg). Control animals in both

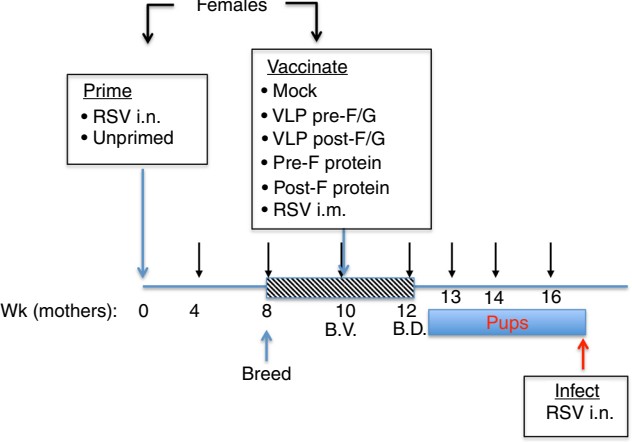

**Fig. 1** VLP immunization of naive and RSV-primed female cotton rats. Diagram of the animal protocol. Female cotton rats were primed by i.n. infection with RSV A/Long (10^5 PFU i.n.) or left unprimed. Eight weeks later, all females were set in breeding pairs with naive males. At week 10 (the time called "before vaccination", B.V.), primed and unprimed females were divided into groups of animals that were mock-immunized, immunized i.m. with different VLPs, F protein preparations, or with live RSV i.m. Pup delivery started on week 12 (time called "before delivery", B.D.). All pups were challenged with RSV at 4 weeks of age, and sacrificed 4 days later

sets were mock-immunized or inoculated i.m. with live RSV, an established positive control for protective RSV immunity[11]. Pups from all females were infected with RSV at 4 weeks of age and then sacrificed 4 days later to assess the protective efficacy and safety of the maternal vaccinations. Four weeks of age was selected for RSV challenge of pups based on the earlier studies

that demonstrated that maternally transferred RSV immunity significantly wanes in cotton rat pups between 1 and 4 weeks after birth, yet remains detectable[11]. Thus, 4 weeks of age is optimal for evaluation of the extent to which maternal immunity to RSV vaccine candidates transferred to pups. Serum samples were obtained from dams at intervals indicated in Supplementary Fig. 5 and 6, and from pups at 4 weeks of age, just before challenge with RSV.

**NA in RSV-primed and naive animals**. RSV NAs in dams were quantified by plaque reduction assay (Figs. 2–3, Supplementary Figs. 5, 6). Before vaccination (B.V., week 10), RSV-primed females consistently showed increased level of RSV NA antibody titers at ~$7.1 \pm 0.2$ Log$_2$ (mean ± standard error (s.e.m.), $n = 55$, Supplementary Fig. 5).

Primed and unprimed females were immunized on week 10 as indicated in Fig. 1 in six different vaccination groups. Supplementary Fig. 6 shows the distribution of RSV NA titers for all female cotton rats (unprimed, back symbols; RSV-primed, red symbols), according to the different vaccination groups. Serum samples were obtained before vaccination (week 10, B.V.) and then ≥2 weeks after vaccination, but <1 week "before delivery" (week 12, B.D.). Additional serum samples from the females were collected on weeks 13, 14, and 16 post-priming.

All animals primed and mock-vaccinated showed a reduction in their RSV NA titers in serum samples taken on week 12 (Fig. 2, Supplementary Fig. 6, Mock, arrow), when compared with serum samples obtained on week 10, supporting our previous observations of a drop in maternal immunity shortly before the time of delivery[6]. Unprimed cotton rats (black bars) strongly responded to all VLP vaccinations, and in the case of Pre-F VLP, produced RSV NA titers similar to those induced in female cotton rats vaccinated with live virus, i.m. (Fig. 2, Supplementary Fig. 6). In primed animals (red bars), VLPs containing the pre-fusion or post-fusion form of the F protein induced higher levels of RSV serum NA before delivery (week 12) than in RSV-primed animals vaccinated with RSV i.m. Importantly, in comparing the RSV NA titers before vaccination (10 wks) vs. NA titers before delivery (12 wks), female cotton rats in all vaccination groups did not exhibit the drop in NA previously seen in RSV-primed, mock-vaccinated animals[6] (Fig. 2 top right panel and Supplementary Fig. 6). Moreover, in the case of VLP and F protein vaccination, we observed a boost of the RSV NA after delivery (12 wks), which was particularly robust and sustained in animals immunized with VLPs containing the pre-F protein (Fig. 2 right panels, Pre-F and Supplementary Fig. 6).

The increased boosting of pre-existent immunity by pre-F VLPs or pre-F protein, however, was transient, exhibiting maximum effect 2–3 weeks after immunization, and subsiding thereafter (Fig. 2 and Supplementary Fig. 6). The strong transient increase in NA induced by pre-F VLPs was not as apparent for post-F VLPs, pre-F, and post-F proteins, or RSV i.m. immunization (Supplementary Fig. 6).

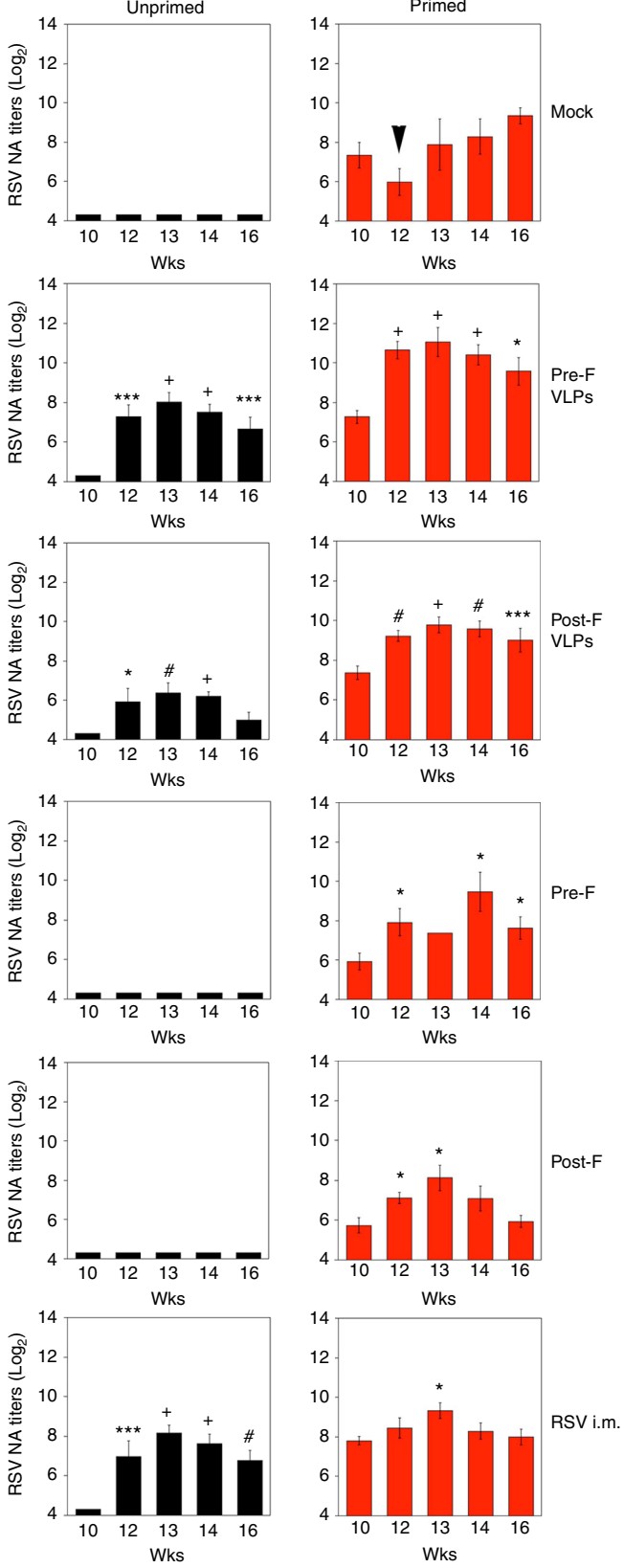

**Fig. 2** RSV serum neutralizing antibodies measured in dams. RSV-primed (right panels, red bars) or unprimed (left panels, black bars) females were bled at the indicated times post-priming. Each bar represent the mean ± s.e.m. of each vaccination group. At each time point, comparison of the NA between the respective mock-vaccinated group and VLPs, F proteins, or RSV-i.m.-vaccinated groups was assessed by one-way ANOVA followed by Tukey post hoc test. *$p < 0.05$, **$p < 0.005$, ***$p < 0.001$, #$p < 0.0005$, +$p < 0.0001$. Week 10 samples represent sera taken before vaccination (B.V.), whereas week 12 samples represent sera taken before delivery (B.D.) of pups

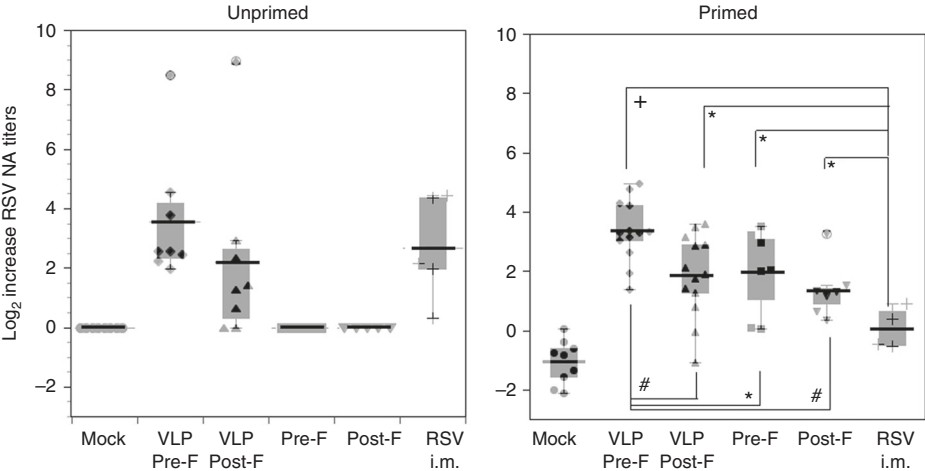

**Fig. 3** Increase in dams' NA. Log$_2$ RSV NA titers in BV and BD samples were compared to determine the change in response ([Log$_2$ titer BD]−[Log$_2$ titer BV]) from each vaccine preparation administered to unprimed or primed dams. Each symbol represents one dam and the line across each group represents mean value. Boxes represent the boundaries of the quartile. Error bars represent s.e.m. Statistically significant boosting in primed animals was assessed by one-way ANOVA followed by Tukey test. *$p < 0.05$; #$p < 0.0005$; +$p < 0.0001$

To quantify the extent of immunity induction/boost by each vaccine, serum RSV NA titers were compared in all females before vaccination (wk 10) and before delivery (Fig. 3). In unprimed animals, vaccination with VLPs led to an increase of mean NA in the female cotton rats of about 2–3 Log$_2$, comparable to that induced by vaccination with RSV i.m., whereas vaccination with soluble F proteins did not. However, in RSV-primed females, those vaccinated with VLPs or purified F proteins displayed a significantly stronger increase of RSV NA titers compared to the group of females vaccinated i.m. with live RSV (ANOVA followed by Tukey post hoc test, +, $p = 0.0001$ for pre-F VLPs). Animals vaccinated with pre-F VLP performed significantly better (ANOVA followed by Tukey ad hoc test, $p < 0.05$) than animals vaccinated with post-F VLPs or purified F proteins. These data indicate that vaccines containing F protein administered during pregnancy can significantly boost NA titers in RSV-primed animals, surpassing live RSV i.m. vaccination in that regard, and that pre-F VLPs may outperform post-F VLPs and F protein subunit vaccines preparations in their boosting ability.

**Total anti-F and anti-G antibody IgG titers**. Maternal anti-pre-F, anti-post-F, and anti-G protein IgG antibodies generated in each of the groups of animals were measured by antigen-specific ELISA assays (Fig. 4 and Supplementary Fig. 7). Unprimed females vaccinated during pregnancy showed strong induction of both anti-pre-F protein and anti-post-F protein antibodies (Fig. 4a, b). As previously reported[10], levels of total pre-F-specific antibodies were slightly lower than levels of post-F-specific antibodies in unprimed animals, even the ones immunized with pre-F. Levels of anti-G protein antibodies were also strongly induced by VLP vaccination, whereas only low levels were detectable in animals immunized i.m. with live RSV (Fig. 4c).

RSV-primed females showed elevated levels of IgG against both, pre-F and post-F targets before vaccination. These animals (Fig. 4d–f) demonstrated a strong boost of total antibodies specific for the pre-F target, post-F target, and G protein. Pre-F VLP immunization resulted in a significantly higher titers of pre-F-specific antibodies than post-F VLP immunization, consistent with the stimulation of pre-F-specific memory responses (Supplementary Fig. 7). In contrast, immunization with the pre-F VLPs and the post-F VLPs resulted in similar levels of antibodies specific for the post-F protein, a result consistent with

previous observations that both pre-F and post-F proteins can induce post-F-specific antibodies.

Importantly, the transient drop in maternal NA titers that occurs shortly before delivery in RSV-primed, mock-vaccinated animals was also seen in both the F-specific and G-specific binding IgG ELISA analysis (Fig. 4d, e, f, arrowheads). Vaccination of pregnant seropositive females with either live RSV i.m. or with any of the VLPs tested prevented this decline in anti-RSV antibody levels in the females before delivery.

There is no difference in the ability of VLPs and soluble F proteins to stimulate memory responses for total anti-F-IgG (Fig. 4g, h). However, VLPs much more efficiently induced total anti-F IgG in naive animals than soluble F protein. Interestingly, there was a clear effect of vaccine format (purified protein vs. VLP) on the ability of pre-F to induce IgG. For example, soluble pre-F administered to unprimed animals induced higher levels of IgG reactive with either pre-F or post-F target, compared to soluble post-F. In the context of VLPs, however, pre-F and post-F were comparable in their ability to induce pre-F and post-F reactive antibodies.

**Serum NA in offspring after maternal vaccination with VLPs**. Cotton rat litters born to unprimed or primed dams vaccinated once with different VLPs, soluble F proteins, or with live RSV administered i.m. were tested for the presence of maternal RSV NA 4 weeks after birth. Litters born to unprimed dams vaccinated with pre-F VLPs showed significant levels of maternal RSV NA remaining in circulation (Fig. 5a, black symbols) compared to animals born to naive, mock-vaccinated dams. Pups born to RSV-primed dams (Fig. 5a, red symbols) vaccinated with pre-F-containing VLPs showed significant enhancement in the remaining maternal NA (~0.7 Log$_2$ increase) compared to animals born to primed, mock-vaccinated dams. Pups born to primed dams vaccinated with RSV i.m., VLPs containing post-F, or soluble pre-F or post-F vaccines did not show significant increase in the remaining levels of RSV NA when compared to primed, mock-vaccinated dams. These data demonstrate that maternal vaccination using pre-F VLPs significantly enhanced the levels of RSV NA in their litter measured at 4 weeks of age, and that the post-F VLPs and the purified proteins were insufficient for that purpose.

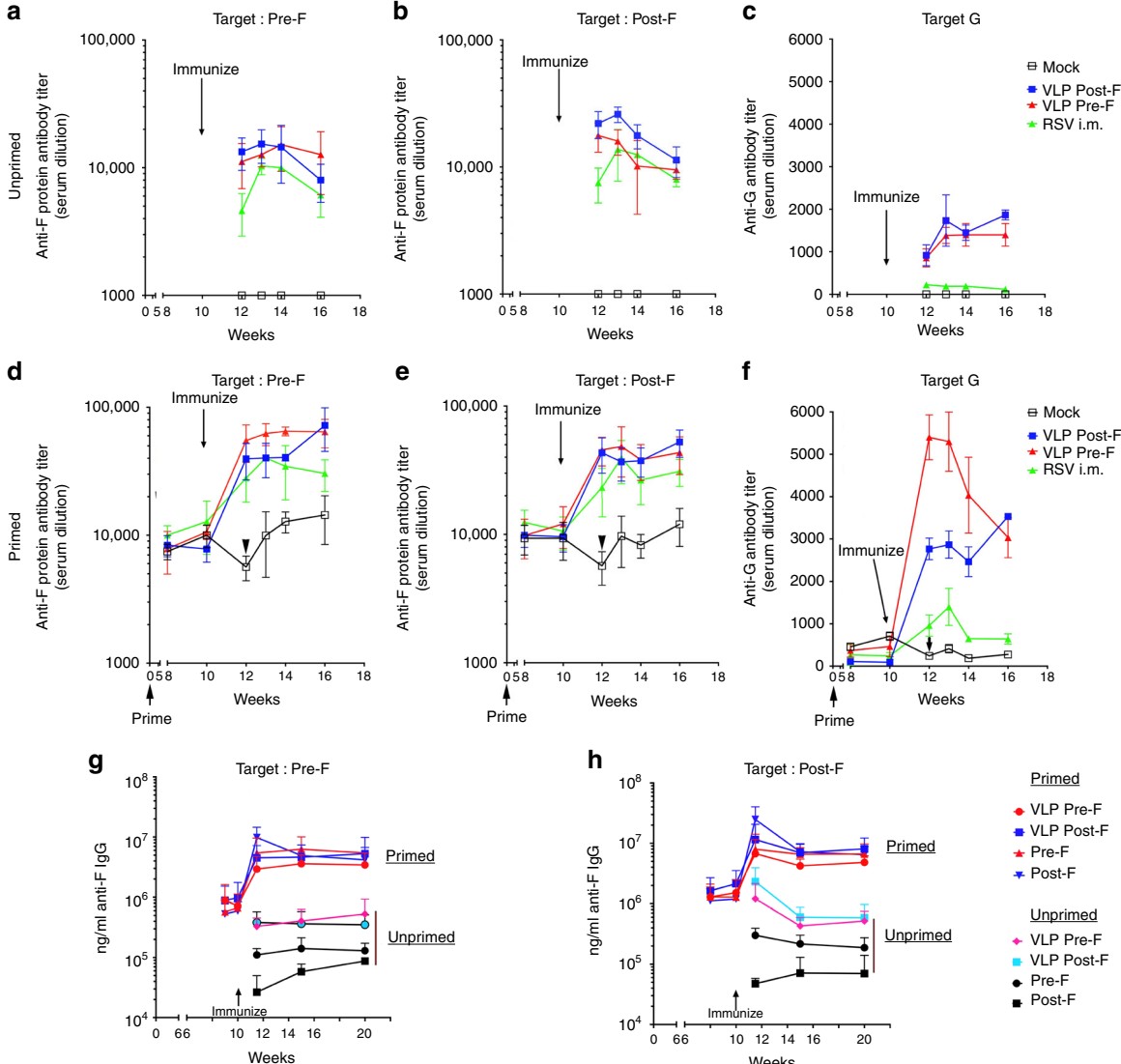

**Fig. 4** Total IgG antibodies specific for RSV pre-F, post-F, and G proteins generated by VLP immunization. Serum samples in each group at each time point during the experiment were pooled. Top row shows total IgG specific for soluble pre-F (**a**), post-F (**b**), or G (**c**) proteins in serum from animals that were unprimed and vaccinated as indicated. Middle row shows total IgG specific for soluble pre-F (**d**), post- F (**e**), or G (**f**) proteins in serum samples from animals that were RSV primed and vaccinated as indicated. Results show the average of 3–5 separate determinations. Arrowheads in mock-vaccinated animals depict the drop in total RSV-specific IgG antibodies measured in serum samples collected before the time of delivery. *p*-values for week 13 serum antibody levels are shown in Supplementary Fig. 7. Bottom row shows total IgG specific for soluble pre-F (**g**), or post-F (**h**) in serum samples from animals that were either RSV unprimed or primed and vaccinated as indicated. Results show the mean ± SD of 3–5 separate ELISA assays using a new pool serum samples from each group

**Protection of offspring from RSV after maternal vaccination.** Litters born to unprimed or primed female cotton rats vaccinated with different VLPs, soluble F proteins, or RSV i.m. were challenged with RSV to evaluate the degree of protection remaining at 4 weeks of age. A small but significant reduction (ANOVA followed by Tukey post hoc test; +p = 0.0001) in viral titers in the lungs was seen in all litters born to unprimed dams vaccinated with VLPs or with RSV i.m. compared to pups of unprimed, mock-vaccinated dams (Fig. 5b, black symbols). However, no differences in protection were seen between groups born to unprimed, mock-vaccinated dams, and litters born to dams vaccinated with purified F proteins. In contrast, vaccination of primed dams with VLPs or purified F proteins significantly improved protection of their litters (Fig. 5b, red symbols) from RSV challenge, while live RSV i.m. immunization did not. In VLP-vaccinated groups, some litters showed full protection of the

lung (Fig. 5b), whereas none of the litters born to dams vaccinated i.m. with the purified F proteins (pre-F or post-F) or with live RSV achieved that level of protection. No significant difference in protection between pups of RSV-primed, pre-F, or post-F VLP-vaccinated dams was observed (Fig. 5b).

Protection of the nose by VLP vaccinations was more modest than protection of the lungs (Fig. 5c). However, two litters born to RSV-primed dams immunized with pre-F VLP had significantly reduced levels of virus in their noses compared to litters born to primed dams immunized with purified pre-F or post-F (Fig. 5c).

**Lung pathology and inflammatory cytokines in litters post challenge.** Next we examined whether vaccination of female cotton rats during pregnancy with VLPs could also benefit the litter by decreasing the lung pathology associated with primary RSV infection. Hematoxylin and eosin-stained lung sections were

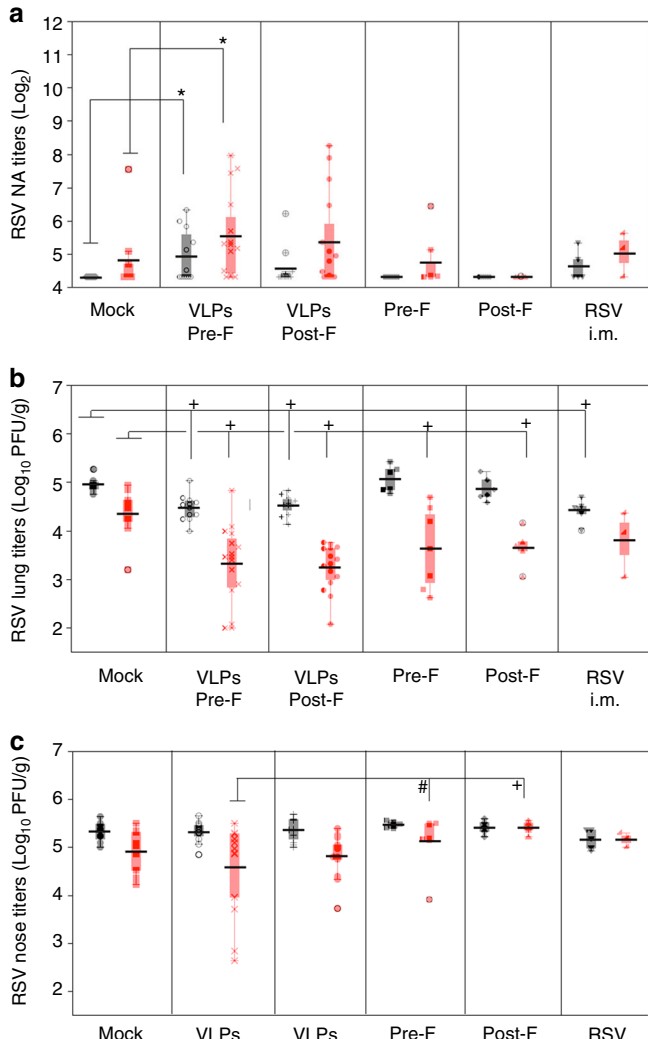

**Fig. 5** Efficacy of protection of offspring by maternal immunization with VLPs. **a** RSV neutralizing antibodies measured in 4-week-old litters derived from the indicated groups of unprimed (black symbols) or RSV-primed (red symbols) dams. Each symbol represents the mean of each litter. Boxes represent the boundaries of the quartiles $Log_2$ NA titers obtained for all litters in a group. Error bars represent s.e.m. Comparison of the responses between the respective mock-vaccinated groups and VLPs or RSV i.m.-vaccinated groups was assessed by non-parametric Kruskal–Wallis test. *$p$ < 0.05. **b** Mean litter lung and **c** nose viral titers in RSV-challenged 4-week-old pups measured on day 4 post infection. Each symbol represents the mean viral titer in each litter. Horizontal lines represent the mean for each group. Boxes represent the boundaries of the quartiles values for each group. Error bars represent s.e.m. Comparison of the protection achieved between the respective mock-vaccinated group and VLPs or RSV-i.m.-vaccinated groups was assessed by one-way ANOVA followed by Tukey post hoc test. +$p$ < 0.0001; #$p$ < 0.0005

scored for RSV-induced lesions as previously described[6]. A significant reduction in peribronchiolitis, a common marker of RSV-related pathology in the cotton rat, was evidenced in animals that were born to primed and vaccinated dams (Fig. 6, red symbols, non-parametric Kruskal–Wallis test). Importantly, animals born to unprimed dams vaccinated with pre-F and post-F VLPs showed a significant reduction in interstitial pneumonia and alveolitis when compared to pups born to unprimed dams that

were mock-vaccinated during pregnancy (Fig. 6, black symbols, non-parametric Kruskal–Wallis test).

Enhanced lung mRNA expression for selected cytokines has been associated with the presence of increased lung pathology in many RSV vaccination and challenge studies. Since we found a reduction of lung pathology in cotton rats born to primed and vaccinated dams, we extended our analysis to determine if expression of these lung cytokines was also reduced. *IL-6*, a cytokine directly induced by viral replication, showed a dramatic decrease in lung expression in pups born to primed and VLPs-vaccinated cotton rat mothers (Fig. 7, red symbols) compared to pups born to mock-vaccinated animals. This result strongly correlates with the reduction in RSV lung viral titer (Fig. 5b) and lung histopathology (Fig. 6) and indicates that vaccination with VLPs during pregnancy not only reduces viral titers but also suppresses lung inflammation and histopathology during RSV infection. Furthermore, expression of *IFN-γ* was also reduced in animals born to primed and VLP-vaccinated dams (Fig. 7, red symbols), showing a greater decrease than seen in pups born to the primed and RSV-i.m.-vaccinated dams. These data demonstrate the safety of vaccination with VLPs during pregnancy in the cotton rat model and corroborate the reduction of lung inflammation in the same vaccinated groups.

## Discussion

Infants under 6 months of age are highly susceptible to RSV infection[2]. However, immunization of this population is problematic for several reasons. First, the immune system of infants is still immature, potentially resulting in an ineffective immune response. Second, the historical failure and dangers of the formalin-inactivated RSV vaccine increase the concern for safety of RSV vaccination of the very young. Third, there have been difficulties in generating live attenuated vaccines with the right balance of immunogenicity and safety for the approval in this population. Fourth, the effects of naturally transferred anti-RSV maternal antibodies on the efficacy of infant vaccination are unknown.

Maternal vaccination could be a safe and effective approach to protect infants from RSV infections. Using serum samples from a cohort of children less than 145 months of age[14], it was recently shown that 100% of infants <1 month of age were seropositive for RSV antibodies, whereas the lowest seroprevalence (25%) was found in infants between 6 and 7 months old. Subsequently, due to natural RSV infection, the percentage of RSV-seropositive infants increased to ~40% by 1 year of age, and reached 100% again by 3 years[14]. Although results of this study might not be generalizable to other infant populations, the authors suggest that the window of RSV susceptibility during infancy has relatively well-defined limits that could be narrowed, at least from one side, by maternal immunization. Thus, a maternal vaccine that could increase and prolong protective antibody levels in the babies could become a routine component of prenatal care.

Several non-replicative RSV vaccine candidates for maternal immunization are currently in clinical trials[15,16]. However, their efficacy in reducing viral replication, extending protection, and reducing disease in the offspring have not been defined or quantified. Surrogates of protection for RSV vaccines during pregnancy are difficult to define since many variables affect the level of protection required for infants at the time of the encounter with virus. The level of maternal antibodies during pregnancy, or more importantly, during the last trimester of pregnancy, has been one of the most important parameters that is indicative of benefit to infants[3]. However, maternal antibodies decay rapidly in infants after birth, and thus their benefit in subsequent months of life depends on the amount and quality

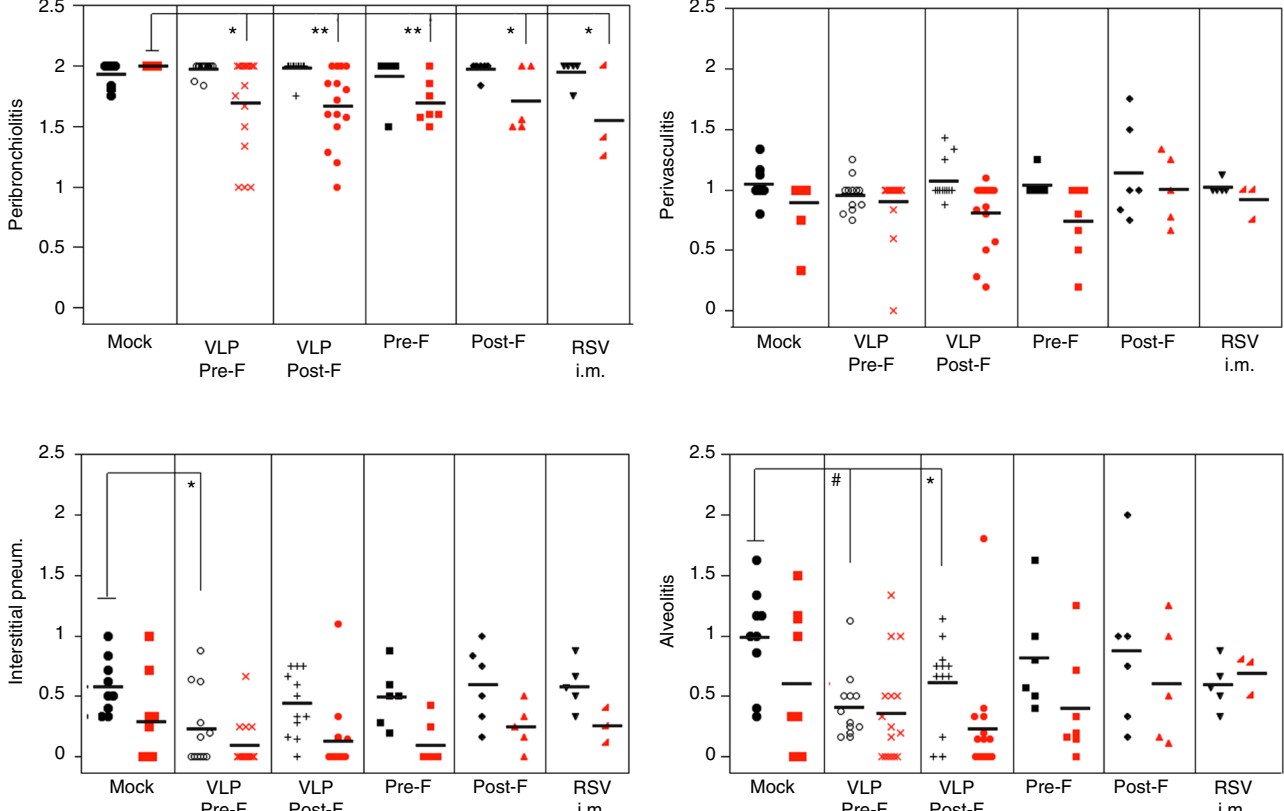

**Fig. 6** Lung pathology in offspring. Lung histopathology scores in RSV-infected litters born to unprimed (black symbols) or RSV-primed (red symbols) dams that were vaccinated once during pregnancy as indicated. Each symbol represents the mean of each litter. Lines across represent mean. Comparison of the histopathology between the respective mock-vaccinated, the test-vaccinated, or RSV i.m.-vaccinated groups was assessed by non-parametric Kruskal–Wallis test. *$p < 0.05$; #$p < 0.005$

received from the mother before delivery or during lactation[17]. In addition, we have reported previously the negative effect that pre-existent RSV immunity has on RSV maternal vaccination[11,18]. Thus, in this study, we explored the boosting capacity of VLP vaccination on pre-existent maternal immunity and its subsequent transfer to offspring.

In cotton rats, we previously defined the threshold period of 4 weeks post birth as the time when protection imparted by RSV-primed dams goes from strong to negligible[11]. The experiments described here were designed based on these previous observations with the purpose of identifying the best vaccine candidate that can extend protection of immunologically naive cotton rat pups past this 4-week threshold.

Our results show that VLPs containing RSV F and G proteins were highly effective at enhancing pre-existing maternal RSV immunity and provided extended protection to progeny with only one immunization during pregnancy. We observed a significant increase in serum NA against RSV in dams only 2 weeks post vaccination in VLPs-vaccinated groups (Figs. 2–3, and Supplementary Fig. 6). This study was performed using a fixed vaccination schedule set at the theoretical mid-term of pregnancy in cotton rats. Further studies will be required to optimize the timing of maternal vaccination with the goal of enhancing protection of the pups from RSV challenge.

The enhancement of RSV NA titers by vaccination of naive females was lower than expected, likely due to the short window of time used for evaluation, 2 weeks, the fact that it was a single immunization, and/or the negative effect of pregnancy on the levels of NA. However, unprimed females were important controls for our study since they demonstrated that VLPs performed better as vaccines in primed animals than in naive animals.

We previously demonstrated that RSV-primed animals undergo a period of reduced RSV immunity that was detected around the time of delivery as evidenced by a drop in NA titers in serum[6]. Now we report that IgG antibodies targeting both forms of F protein and the G protein showed a decrease in titer in these females, indicating that this drop is a generalized phenomenon that is present in these animals before delivery. Although this finding has not yet been corroborated in humans, there is recent evidence that pregnancy is a time of increased RSV vulnerability[4,5]. However, we demonstrate that vaccination during pregnancy with RSV i.m. or with either of the two different F-G RSV VLP preparations overcome the transient immunosuppression that occurs shortly before delivery. We hypothesize that vaccination during pregnancy could reload and/or maintain the pool of maternal anti-RSV antibodies, thus benefiting the mothers and also increasing the levels of antibodies transferred to the fetus through the placenta and/or through nursing.

Pre-F-containing VLPs were found to be significantly more efficient than the post-F VLPs vaccine or the purified F protein vaccines in boosting pre-existent RSV NA in primed female cotton rat (Fig. 3). Pre-F VLPs also enhanced the levels of NA transferred to litters, in addition to the protection of the lung from RSV replication (Fig. 5b). Furthermore, the protection of the nose in the challenged pups of primed dams vaccinated with pre-F VLPs showed improvement in the protection of several different litters (Fig. 5c). Although this result did not achieve statistical significance when litters from primed, mock-vaccinated animals were compared, the improvement was significant when compared to litters from dams primed and vaccinated with the purified F protein preparations. This result could certainly depend on the pre-existent repertoire of B and T cells to be re-

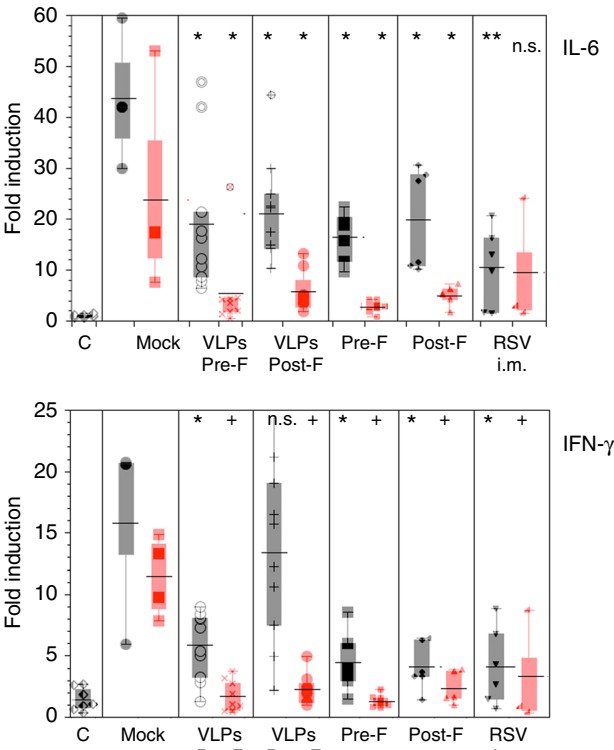

**Fig. 7** Lung cytokine expression. Expression of mRNA for the cytokines IL-6 and IFN-γ in the lungs of RSV-infected litters born to naive (black symbols) or RSV-primed (red symbols) dams vaccinated as indicated ($n = 3$–10). Control (C) represents animals that were naive and not infected (used as baseline for mRNA expression). Symbols represent the mean of individual litters. Bar across represent the mean ± s.e.m. of a group. Boxes represent the boundaries of the quartiles. Comparison of the expression between the respective mock-vaccinated group and VLPs or RSV-i.m.-vaccinated groups was assessed by one-way ANOVA followed by Tukey post hoc test. $+p < 0.0001$; $**p < 0.005$; $*p < 0.05$

stimulated by the vaccine. The reasons why the difference in immune responses in dams to the pre-F and the post-F was not significantly translated into an increased transfer of NA and differences in the lung protection in offspring are unknown and will be the subject of future investigations.

Human studies have estimated the half-life of maternally derived anti-RSV antibodies to be between 28 and 80 days, depending on the study[19–22], whereas the half life of cotton rat antibodies is ~7 days[18]. Since in this study we tested NA titers in cotton rats at 4 weeks of age, theoretically four half-lives passed by the time of challenge. The levels of RSV NA antibodies in pups of primed dams vaccinated with VLPs at 4 weeks (~5.5 $Log_2$) were significantly higher than those in pups of primed, mock-vaccinated dams, suggesting that in humans these VLP vaccines could potentially elicit enhanced and uniform protection of babies beyond the 6th month.

Overall our data demonstrate that RSV F/G VLP vaccination of cotton rats during pregnancy robustly stimulates pre-existent RSV immunity with only one immunization, provides enhanced protection of offspring, and reduces pulmonary inflammation in the pups normally associated with RSV infection. Comparison of vaccination with VLPs vs. equivalent concentrations of the purified F show superiority under the conditions tested and further emphasize the potential of VLP platform to improve vaccine efficacy.

## Methods

**VLP preparation and characterization**. VLPs containing the RSV F and G proteins were formed with the NDV core proteins NP and M[23,24]. The cDNAs encoding the NDV NP and M protein have been previously described[25]. The RSV G and F proteins were incorporated into these VLPs by constructing chimera protein genes composed of ectodomains of the G or F glycoproteins fused to the transmembrane and cytoplasmic domains of the NDV HN protein or NDV F glycoprotein, respectively, to generate H/G and F/F chimera proteins. The NDV domains specifically interact with the NDV NP and M protein resulting in efficient incorporation of the chimera proteins into VLPs. The construction and expression of the chimera protein NDVHN/ RSVG (H/G), the stabilized chimera pre-fusion F protein (DS Cav1 pre-F/F), and the stabilized chimera post-fusion F protein (post-F/F) have been previously described[9].

For preparations of VLPs to be used as immunogens (VLP-H/G + Pre-F/F, VLP-H/G + Post-F/F), ELL-0 cells (ATCC CRL-12203) growing in T-150 flasks were transfected with cDNAs encoding the NDV M protein, NP, the chimeric proteins H/G, and either Pre-F/F or Post-F/F as previously described[12,24]. At 24 h post-transfection, heparin (Sigma, H4784) was added to the cells at a final concentration of 10 μg ml[−1][11] to inhibit re-binding of released VLPs to cells. At 72, 96, and 120 h post-transfection, cell supernatants were collected and VLPs purified by sequential pelleting and sucrose gradient fractionation as previously described[8,12,24]. Concentrations of proteins in the purified VLPs were determined by silver-stained polyacrylamide gels and by western blot analysis using marker proteins for standard curves[8,24]. The conformation of F protein in the VLP preparations was verified by reactivity to mAbs as previously described[8,24].

**Preparation of soluble F and G proteins**. The constructions of genes encoding the soluble pre-F protein, the soluble post-F protein, and the soluble G protein used for target in ELISA were previously described[9]. For preparation of these soluble proteins, expi293F cells (ThermoFisher A14527) were transfected with pCAGGS vector containing sequences encoding the soluble proteins. At 5–6 days post transfection, total cell supernatants were collected and cell debris removed by centrifugation. Pre-fusion F, post-fusion F, and G polypeptides were then purified on columns using the His tag and then the strep tag as previously described[26].

**ELISA protocols**. For ELISA, wells of microtiter plates (ThermoFisher/Costar 2797) were coated with either purified soluble pre-fusion F protein, soluble post-fusion F protein, or soluble G protein and incubated for 24 h at 4 °C. Wells were then incubated in PBS-2% BSA for 16 h. Different dilutions of sera, in 0.05% Tween and 2% BSA, were added to each well and incubated for 2 h at room temperature. After six washes in PBS, sheep anti-mouse antibody coupled to HRP (Sigma A5906) was added in 50 μl PBS-2% BSA and incubated for 1.5 h at room temperature. Bound HRP was detected by adding 50 μl TMB (3,3′5,5′-tetra-methylbenzidin, ThermoFisher34028) and incubating for 5–20 min at room temperature until blue color developed. The reaction was stopped with 50 μl 2N sulfuric acid. Color was read in SpectraMax Plus Plate Reader (Molecular Devices) using SoftMax Pro software. ELISA titers were defined as the dilution of sera that resulted in an optical density of 0.4, which was at least 20-fold over background as in Fig. 3a–f. Results shown in panels g and h are ng ml[−1] IgG calculated using a standard curve of purified cotton rat IgG.

**Animals**. *Sigmodon hispidus* cotton rats were obtained from an inbred colony maintained at Sigmovir Biosystems, Inc. (Rockville, MD). Three-week-old female cotton rats were recruited for these studies, and randomly tagged and separated in groups as indicated (Supplementary Table 1). Eight weeks later, females were paired with males ~2 weeks older than the females for mating. Animals were pre-bled before inclusion in the study to rule out the possibility of pre-existent antibodies against RSV. The colony was monitored for antibodies to paramyxoviruses and rodent viruses, and no such antibodies were found. All studies were conducted under applicable laws and guidelines and after approval from the Sigmovir Biosystems, Inc. Institutional Animal Care and Use Committee. Animals were housed in large polycarbonate cages and fed a standard diet of rodent chow and water ad libitum. All cotton rats born as a result of breeding during these studies were used for RSV infection at 4 weeks of age and are referred to as "pups".

**Viruses and viral assays**. The prototype Long strain of RSV was obtained from American Type Culture Collection (ATCC VR-26, Manassas, VA). Virus was propagated in HEp-2 cells (ATCC CCL-23) due to their permissiveness, and serially plaque-purified to reduce defective-interfering particles. The single pool of virus containing $10^{7.6}$ PFU/ml was used for all experiments. To adjust the dose, stock aliquots were diluted with PBS for intranasal (i.n.) infections. Viral titers in the lungs and in the nose of RSV-infected pups were determined as described elsewhere[11] and normalized for the weight and expressed as PFU gm[−1] of tissue.

**RSV NA assay**. RSV NA titers were measured by 60% plaque reduction assay as previously described[27]. In brief, 4-fold or 6-fold dilutions of heat-inactivated serum samples were incubated with RSV/A/Long strain and inoculated onto HEp-2 cells. After 4 days of incubation under 0.75% methylcellulose overlay, the overlay was removed and the cells were fixed and stained with crystal violet. Plaques were

quantified and reciprocal NA titers were determined as previously described (the limit of detection of this assay is 4.32 $Log_2$)[11].

**Cytokine gene expression by real-time RT-PCR.** Total RNA was extracted from homogenized lung tissue using the RNeasy purification kit (QIAGEN). One µg of total RNA was used to prepare cDNA in a volume of 20 µl (QuantiTect Reverse Transcription Kit from Qiagen). cDNA was diluted to 10 µg ml$^{-1}$ and 3 µl were used for each 25 µl real-time PCR reaction (QuantiFast SYBR Green PCR Kit from Qiagen) with final primer concentrations of 0.5 µM. Primer sequences for β-actin, IFN-γ, and IL-6 are provided in Supplementary Table 2. Reactions were set up in 96-well plates and amplifications were performed on a Bio-Rad iCycler (MyiQ Single Color). Delta Ct method was used to calculate relative gene expressions that were normalized to β-actin as a housekeeping gene[6,11].

**Lung histopathology.** Lungs were dissected and inflated with 10% neutral buffered formalin to their normal volume, and then immersed in the same fixative solution. After fixation, lungs were embedded in paraffin, sectioned, and stained with hematoxylin and eosin (H&E). An average pathology score was determined for each group based on four parameters of pulmonary inflammation: peribronchiolitis (inflammatory cell infiltration around the bronchioles), perivasculitis (inflammatory cell infiltration around the small blood vessels), interstitial pneumonia (inflammatory cell infiltration and thickening of alveolar walls), and alveolitis (cells within the alveolar spaces). Slides were scored blindly on a 0–4 severity scale as previously described[6].

**Experimental design.** Female cotton rats were bled and a subgroup of all female used were primed by RSV A/Long infection using a dose of 10$^5$ PFU/animal (50 µl per animal intranasally (i.n.)) (Fig. 1, Supplementary Table 1). At week 8, all females were set up in breeding pairs with RSV-seronegative males. Females were bled for serum collection at 4, 8, and 10 weeks post-priming. At week 10 (the time generally corresponding to the middle of pregnancy) females were separated into six additional groups and vaccinated i.m. with TNE (50 mM Tris-HCl, pH 7.4, 150 mM NaCl, 5 mM EDTA), VLPs containing the attachment RSV G protein and a pre-fusion or a post-fusion form of the F protein (VLP Pre-F/G and VLP Post-F/G, respectively); 100 µg of VLPs per animal and containing 10 µg of F protein and 10 µg of G protein per dose, purified pre-F protein (10 µg per animal), purified post-F protein (10 µg per animal), or a control group vaccinated i.m. with live RSV A/Long (10$^5$ PFU RSV per animal)[11]. Females started delivering pups after the week 12 post-priming. All pups were eye-bled, and challenged with RSV A/Long i.n. (10$^5$ PFU per animal) at 4 weeks of age. On day 4 post infection, all pups were sacrificed for analysis of nose and lung viral titers, lung histopathology, and expression of mRNA corresponding to select Th1 and Th2 cytokines as previously described[28]. An additional group of five females remained unprimed and unvaccinated. This group of females was used as a source of pups that remained uninfected, and provided basal levels of lung gene expression for comparison. Serum samples were obtained from pups before RSV challenge. Dams were kept on the study for ~5 weeks after delivery of pups for additional serum collection. The results presented are collected from three independent experiments (XV-146, XV-176, and XV-186).

**Statistical analyses.** For determination of serum NA titers in each group of female cotton rats, mean ± s.e.m. was determine. Comparison of the indicated groups was performed by one-way ANOVA followed by Tukey post hoc test. For litters, values of viral titers, serum NA, cytokine mRNA expression, and histopathology of pups in a litter were averaged and their mean ± s.e.m. were plotted. Comparison between indicated groups was performed by one-way ANOVA followed by Tukey post hoc test or non-parametric Kruskal–Wallis test, as indicated. For analysis, KaleidaGraph 4.5 and GraphPad Prism 7.0c for Mac OS X were used.

**Data availability.** All data relevant to this study are available from the corresponding author upon request.

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

## Acknowledgements

This work was supported by NIH grants AI109926 (J.C.G.B.), AI114809 (T.G.M.), and by a Grant from the Charles H. Hood Foundation to T.G.M. The authors would like to thank Dr. Stefanie N. Vogel for suggestions and edition of the final manuscript. We

would like to acknowledge Ms. Martha Malache, Mr. Charles Smith, Mr. Fredy Rivera, Ms. Ana Rivera, and Mr. Arash Kamali for their technical support with the animals. Also we would like to acknowledge Ms. Zenab Chinmoun, Mr. Daniel Stylos, and Dr. Wei Zhang for their laboratory support.

## Author contributions

J.C.G.B., T.G.M., and M.S.B. planned the experiments and interpreted the data. L.M.P., R. O.O., and M.C.P. performed all the experiments with the exception of the soluble proteins' vaccination experiment. L.M. prepared the vaccine preparations and performed in vitro characterization of the VLPs and soluble F proteins. J.C.G.B. and L.R.F. performed the soluble proteins' vaccination experiment. J.C.G.B., T.G.M., and M.S.B. wrote and edited the paper.

## Additional information

**Competing interests:** J.C.G.B., L.M.P., R.O.O., M.C.P., L.R.F., M.S.B. from Sigmovir Biosystems Inc. (SBI) declare no competing interests. SBI is a contract research organization and perform vaccine and therapeutic studies on RSV using the cotton rat model. T.G.M., L.M., and the University of Massachusetts Medical School (UMASS) have a patent on use of NDV VLPs as vaccine platforms. They have also pending patent applications on incorporation of RSV glycoproteins into VLPs.

