## [Peer Review File · Nature Communications]

Reviewers' comments:

Reviewer #1 (Remarks to the Author):

This paper is an addition to previous publications by the same groups in which Newcastle disease virus (NDV) like particles (VLPs) incorporating the ectodomains of the Respiratory syncytial virus (RSV) F and G glycoproteins have been used to immunize cotton rats. These VLPs have already demonstrated their efficacy to induce high levels of neutralizing antibodies that conferred protection against a RSV challenge.

This type of RSV F and G glycoprotein containing VLPs has been used in the current manuscript to boost pre-existing RSV immunity in pregnant cotton rats and evaluate their protective effect in pups four weeks after birth. The results obtained showed a modest reduction (about 10 fold) of RSV titers in the lungs four days after challenge of animals born from pre-infected mothers boosted with VLPs during pregnancy compared with those born from pre-infected but mock-boosted controls. This is perhaps a promising starting point for development of a RSV vaccine to be used for maternal vaccination, a strategy being considered as a suitable alternative to RSV vaccination very early in life when safety concerns and immune response failures are of utmost importance.

In my opinion, the VLP approach needs a straight forward comparison with other alternatives, particularly subunit vaccines made with highly purified glycoproteins, to really appreciate its applicability. Subunit vaccines may have the advantage of being very homogeneous preparations of well characterized glycoproteins while VLPs may benefit from the presentation to the immune system of repeating antigenic structures on their surfaces. These two properties, homogeneity and immunogenicity, will impact the future of any candidate vaccine and require careful consideration. It is therefore of additional concern the lack of characterization of the VLPs used in this study. Granted that the authors have already reported characterization (mostly by western blot) of the VLPs used in previous studies, but I think that at least total protein profile by SDS-PAGE and reactivity with pre- and post-fusion F monoclonal antibodies of the VLPs used in the current study should be included in the manuscript.

Other points to be considered by authors:

1. Figures are not numbered
2. Title and line 104: The term RSV virus-like particle (VLP) is misleading. The authors have used NDV, not RSV, VLPs which incorporate the ectodomains of RSV glycoproteins
3. Line 64: "Maternal vaccination against many viral infections..." Remove, many. There are not so many licensed vaccines for maternal vaccination.
4. Line 72: The mortality rates reported in the cited reference are ten times lower than indicated.
5. Since G glycoprotein seems to be poorly immunogenic, is there a reason for being included in the VLPs?
6. Whereas Pre-F containing VLPs induced slightly higher levels of neutralizing antibodies than Post-F VLPs in RSV primed mothers, pups born from the latest mothers had the highest neutralizing antibody titers. Any explanation for this apparent discrepancy or it is just an inconsistency (even if statistically significant) due to the modest protective effects seen throughout the study? Following this argument, is the very modest protective effect in the nose of pups from mothers boosted with 100 µg of Pre-F VLPs but not with 75 µg of Pre-

F VLPs statistically significant?
7. line 527: "expression"??

Reviewer #2 (Remarks to the Author):

Manuscript#: NCOMMS-17-15823

Title: Efficacy of RSV virus-like particle (VLP) vaccine candidates in naïve and previously infected cotton rat mothers and their offspring

Key results

The study assesses immunogenicity and efficacy of novel VLP vaccine candidates composed of Newcastle disease virus core NP, M, and ectodomains of RSV fusion (F) and attachment (G) proteins using cotton rats (CR) to model maternal immunization. Results show immunization of RSV-primed CRs with RSV F/G VLP vaccines during gestation was highly effective at boosting maternal antibody with only one immunization, provided extended protection to pups, and reduced pulmonary inflammation otherwise associated with RSV infection.

Validity

The manuscript does not have flaws significant enough to prohibit publication, but suggestions for improvement are provided in this review.

Originality and significance

Evaluation of vaccine safety and efficacy in relevant preclinical models is an important component of the rationale for advancing vaccine candidates into the clinic, particularly prior to entering sensitive pregnant women and infant populations. Cotton rats are susceptible to RSV infection, and became a standard preclinical model in the 1980s when shown to recapitulate some features of vaccine enhanced disease seen in RSV-naïve infants following subsequent infection with RSV. Several of the authors have long experience with the CR model, and have previously published on optimizing the model to evaluate RSV vaccine candidates using a maternal immunization strategy. Testing in RSV-primed animals more accurately represents the human context, as adult vaccine targets have been primed by natural infection with RSV.

Data and methodology

The general study approach used is valid, and though suggestions to improve study design (thereby improving data quality) and analyses are included in this review, they are unlikely to significantly alter key conclusions. The report and referenced publications provide sufficient detail to enable reproducing the results. The clarity/accuracy of the study write up could be improved, and suggestions are provided.

Suggestions for improving clarity/accuracy in the study write up:

- Consistent with a report that describes a preclinical study, suggest referring to cotton rat

females or dams vaccinated or immunized during pregnancy or gestation, rather than vaccination or immunization of **mothers**. Alternately, note they are **cotton rat** mothers.

- Abstract-

- o Line 46: Suggest being more specific that maternal **antibody** (rather than maternal immunity), is protective against RSV infection in infants early in life.
- o Line 58: VLP immunization of **cotton rats** during pregnancy (or gestation) provided (rather than induced) significant protection of naïve pups from RSV challenge 4 weeks after birth and reduced pulmonary inflammation otherwise associated...
- o Line 60: **Results in the cotton rat maternal immunization model indicate a** VLP vaccine with RSV F and G proteins **could be** a safe and effective RSV vaccine for maternal and adult vaccination.

- Introduction-

- o Line 86: Immunity (or Ab) from natural infection with RSV might be better described as **incompletely**, rather than poorly protective.
- o Line 109: ...efficacy of **these vaccines in providing** protection from RSV challenge to offspring.
- o Line 113: We report that VLP immunization significantly **increased neutralizing antibody titers** (rather than benefits) in both RSV-seropositive pregnant females and their offspring. (Subsequent text describes the other "benefits", but does not specifically mention neutralizing antibody.)

- Results:

- o Line 143: ...then sacrificed 4 days later **to assess the protective efficacy and safety of the VLP vaccines**.
- o Line 291: These data clearly demonstrate the safety of vaccination with VLPs during pregnancy **in a cotton rat model** and corroborate the strong reduction of lung inflammation in the same vaccinated groups.

- Discussion conclusions:

- o Line 301: Furthermore this type of vaccine may not be readily approved for **young** infants.
- o Line 302: Fourth, effects of **naturally** transferred anti-RSV maternal antibodies on **active infant** vaccination efficacy are unknown.
- o Line 319: It may be true that testing of other clinical stage RSV vaccines in RSV-seropositive pregnant animals has not been published, but the authors cannot claim that it has not been done. Modify your statement or delete.
 - RSV Vaccines for the World (2013), Poster 113 'Immunogenicity, efficacy and safety of a respiratory syncytial virus recombinant F protein vaccine in cotton rats' Ann-Muriel Steff et al. GlaxoSmithKline Vaccines, Laval, Quebec, Canada
 - RSV Vaccines for the World (2013), Poster 114 'Proof of concept of the efficacy of a maternal RSV, recombinant F protein, vaccine for protection of offspring in the guinea pig model' Ann-Muriel Steff et al. GlaxoSmithKline Vaccines, Laval, Quebec, Canada
 - 8th Vaccine and ISV Congress (2014), Poster 1.64 "A cotton rat model for evaluation of a

respiratory syncytial virus vaccine for pregnant women" A. Woods*, K. Hashey, J. Monroe, K. Friedrich, P. Dormitzer, C. Shaw, Novartis Vaccines, Inc., USA

o Line 350: The authors comment on the low NA titers following VLP vaccination of naïve females. While acknowledging that pregnancy may have an impact, it is not unusual to see only low level Ab responses to a first (priming) immunization in naïve animals, with significant Ab boosting resulting from a second immunization (Plotkin, 2012 Vaccines 6th edition, Saunders, Chapter 2, Vaccine Immunology by Claire-Anne Siegrist). Consider revising discussion accordingly, e.g. Induction of lower NA titers were elicited following VLP vaccination of naïve CRs, consistent with a priming, rather than boosting immunization, as seen in the RSV-primed females...

o Line 387: Overall, our data demonstrate that RSV F/G VLP vaccination of **cotton rats** during pregnancy robustly stimulates pre-existing RSV immunity **with only one immunization, provided** enhanced protection of offspring, **and reduced pulmonary inflammation otherwise associated with RSV infection.**

Appropriateness of statistics and treatment of uncertainties

- General study design: Please clarify why the mock and RSV i.m. group sizes are smaller than the VLP group sizes. Better power for statistical comparisons could have been achieved if mock and RSV i.m. group sizes were equal to the rest.
- Please state what statistical software was used for analyses.
- Please consistently refer to either SE or SEM throughout report.
- Figure 1A would be improved by labeling the week 13, 14, 16 bleeds in the mothers.

Lines 526-529 / Figure 1B:

- Suggest that the text be revised to clarify that "At each time point, comparison of the expression between..."
- Based on Supplemental Figure 3, it appears that one of four animals in the primed mock group has particularly high NA titer values. The non-parametric Kruskal-Wallis test is more robust to outliers and may be useful to assess sensitivity of the results to inclusion of this animal.
- Did the authors consider other methods (vs Newman-Keuls) for post hoc testing of pairwise comparisons? For example, the Tukey method is more robust for unequal group sizes and could provide better Type 1 error control for comparisons among 5 groups. Newman-Keuls may lead to anti-conservative inference in some scenarios (p values smaller than they should be). What is the rationale for using Newman-Keuls post hoc testing? (ref: <https://www.ncbi.nlm.nih.gov/pubmed/22420233>)

Lines 531-537 / Figure 1C:

- Same comment/question as above for choice of Newman-Keuls vs other methods.
- Suggestion to add error bars to plot.
- Why is red color used for markers of RSV i.m.?
- The p value given in the text on line 196 ($p < 0.005$) differs from that shown in the legend for Figure 1C ($p < 0.0005$). Please correct as appropriate.

Lines 538-547 / Figure 2 / Supplementary Fig 4:

- Please define what Figure 2 error bars represent.

- Please clarify what statistical methods were used to generate the p values in Supp Fig 4. Were data log-transformed for analysis? They are shown as log₁₀ titers in Figure 2, and ng/ml in Supp Fig 4. Additionally, samples were pooled by group and assayed multiple times to generate the data presented. The resulting very small group sizes (3-5 assay runs contributing to each mean), warrant an assessment of whether one-way ANOVA assumptions are met. Consider using Kruskal-Wallis test as a non-parametric equivalent to one-way ANOVA as either the primary analysis or as a sensitivity analysis.

Lines 549-561 / Figure 3

- Please clarify how litter effects were handled in statistical analyses. It appears that the analysis was conducted with the pup as the experimental unit of analysis when in fact it should be the litter. Using the pup as the statistical unit of measure for this study design will generally inflate the Type 1 error rate, leading to anti-conservative inference (p values smaller than they should be). Use of a mixed effects model (to account for correlation among pups in the same litter) would be ideal. Calculating the mean of animals within the same litter, than conducting ANOVA (or similar method) on the litter means would be acceptable as well.

(ref: <https://www.ncbi.nlm.nih.gov/pubmed/23522086>)

- Scatter plots with SE bars are preferable for Fig 3A, B, C.

Lines 563-578 / Figure 4

- Figure 4A. Same issue as above: it is problematic to use the pups rather than the litters as the experimental use of analysis. In addition, pathology score is an ordinal discrete measure, and ANOVA, while robust to departures from normality for large group sizes, may not be so for these small group sizes. Recommend using a non-parametric test instead.

- Figure 4B. Same issue with using pups as the statistical unit of analysis. Additionally, this plot could be improved by using a scatter plot (with mean & SE bars) rather than bar chart. This would allow the reader to evaluate the distribution of the data and appropriateness of statistical methods that assume normality.

References

The references cited are appropriate with the following suggestions/requests.

- Consider replacing Nair reference now that the updated RSV global burden data has been published. (Shi et al. Lancet in press, available online July 7, 2017)

[http://dx.doi.org/10.1016/S0140-6736\(17\)30938-8](http://dx.doi.org/10.1016/S0140-6736(17)30938-8)

- Line 73: Provide reference(s) that RSV maternal Ab likely protects infants in the first months of life.

- Reference 10 is incomplete. Please update with publication information.

Responses to the Reviewers.

I would like to thank the Reviewers for their kind words and suggestions that have greatly improved the quality of our work. We have performed two additional experiments in order to respond to the reviewers' concerns and suggestions. Due to the time required for these additional studies (6 month), our resubmission has been considerably delayed.

It is important to add that for a complete comparison between groups, we have removed from the analysis the group vaccinated with pre-F VLPs at low concentrations (75 µg), and thus, we have compared only the groups vaccinated with 100 µg VLPs along with the additionally requested vaccination groups containing purified F proteins in the pre- or post-fusion conformations.

The following are our responses to each of the Reviewers' comments.

Reviewer #1

In my opinion, the VLP approach needs a straight forward comparison with other alternatives, particularly subunit vaccines made with highly purified glycoproteins, to really appreciate its applicability.

A new experiment in which the VLP vaccines were compared with preparations of subunit vaccines containing the F protein of RSV in the pre- and post- fusion conformations was performed and the data are now included in the revised manuscript.

It is therefore of additional concern the lack of characterization of the VLPs used in this study. Granted that the authors have already reported characterization (mostly by western blot) of the VLPs used in previous studies, but I think that at least total protein profile by SDS-PAGE and reactivity with pre- and post-fusion F monoclonal antibodies of the VLPs used in the current study should be included in the manuscript.

The data on the characterization of the VLP F and G antigens as well as the soluble F protein preparations used for the additional

studies are now provided in new Supplementary Figures 1 to 4.

Other points to be considered by authors:

1. Figures are not numbered

Figure 1 is now numbered.

2. Title and line 104: The term RSV virus-like particle (VLP) is misleading. The authors have used NDV, not RSV, VLPs which incorporate the ectodomains of RSV glycoproteins

The title has been changed accordingly.

3. Line 64: "Maternal vaccination against many viral infections...." Remove, many. There are not so many licensed vaccines for maternal vaccination.

"Many" has been removed from the statement.

4. Line 72: The mortality rates reported in the cited reference are ten times lower than indicated.

Mortality rates have been updated. We apologize for the mistake.

5. Since G glycoprotein seems to be poorly immunogenic, is there a reason for being included in the VLPs?

It has been demonstrated by numerous investigators that antibody to the G protein has an important role in protection from disease. Thus, we believe that there is an improvement on the efficacy of the vaccine by including the G protein. In addition to including new neutralizing epitopes, the G protein could affect the conformation of F and vice versa. This is currently under investigation.

6. Whereas Pre-F-containing VLPs induced slightly higher levels of neutralizing antibodies than Post-F VLPs in RSV primed mothers, pups born from the latest mothers had the highest neutralizing antibody titers. Any explanation for this apparent discrepancy or it is just an inconsistency (even if

statistically significant) due to the modest protective effects seen throughout the study?

After performing two additional experiments comparing these groups, pre-F-expressing VLPs induced higher neutralizing antibodies in mothers and also in their litters, so the previous result was affected by lower sample numbers. In addition, now we are comparing litters to strengthen our statistical analysis.

7. Following this argument, is the very modest protective effect in the nose of pups from mothers boosted with 100 µg of Pre-F VLPs but not with 75 µg of Pre-F VLPs statistically significant?

The 75 microgram data has been removed from the manuscript. The effects of different concentrations of VLPs on protective immune responses is currently under investigation.

8. line 527: "expression"??

The words "gene expression" were replaced by RNA

Reviewer #2 (Remarks to the Author):

Manuscript#: NCOMMS-17-15823

Title: Efficacy of RSV virus-like particle (VLP) vaccine candidates in naïve and previously infected cotton rat mothers and their offspring

Key results

The study assesses immunogenicity and efficacy of novel VLP vaccine candidates composed of Newcastle disease virus core NP, M, and ectodomains of RSV fusion (F) and attachment (G) proteins using cotton rats (CR) to model maternal immunization. Results show immunization of RSV-primed CRs with RSV F/G VLP vaccines during gestation was highly effective at boosting maternal antibody with only one immunization, provided extended protection to pups, and reduced pulmonary

inflammation otherwise associated with RSV infection.

Validity

The manuscript does not have flaws significant enough to prohibit publication, but suggestions for improvement are provided in this review.

Originality and significance

Evaluation of vaccine safety and efficacy in relevant preclinical models is an important component of the rationale for advancing vaccine candidates into the clinic, particularly prior to entering sensitive pregnant women and infant populations. Cotton rats are susceptible to RSV infection, and became a standard preclinical model in the 1980s when shown to recapitulate some features of vaccine enhanced disease seen in RSV-naïve infants following subsequent infection with RSV. Several of the authors have long experience with the CR model, and have previously published on optimizing the model to evaluate RSV vaccine candidates using a maternal immunization strategy. Testing in RSV-primed animals more accurately represents the human context, as adult vaccine targets have been primed by natural infection with RSV.

Data and methodology

The general study approach used is valid, and though suggestions to improve study design (thereby improving data quality) and analyses are included in this review, they are unlikely to significantly alter key conclusions. The report and referenced publications provide sufficient detail to enable reproducing the results. The clarity/accuracy of the study write up could be improved, and suggestions are provided.

Suggestions for improving clarity/accuracy in the study write up:

- Consistent with a report that describes a preclinical study, suggest referring to cotton rat females or dams vaccinated or immunized during pregnancy or gestation, rather than vaccination or immunization of **mothers**. Alternately, note they are **cotton rat** mothers.

These changes have been implemented throughout the manuscript.

- Abstract-

- o Line 46: Suggest being more specific that maternal **antibody** (rather than maternal immunity), is protective against RSV infection in infants early in life.

It was corrected as requested.

- o Line 58: VLP immunization of **cotton rats during pregnancy (or gestation) provided** (rather than induced) significant protection of naïve pups from RSV challenge 4 weeks after birth and reduced pulmonary inflammation otherwise associated...

- o Line 60: **Results in the cotton rat maternal immunization model indicate a** VLP vaccine with RSV F and G proteins **could be** a safe and effective RSV vaccine for maternal and adult vaccination.

It was corrected as requested.

- Introduction-

- o Line 86: Immunity (or Ab) from natural infection with RSV might be better described as **incompletely**, rather than poorly protective.

It was corrected as requested.

- o Line 109: ...efficacy of **these vaccines in providing** protection from RSV challenge to offspring.

It was corrected as requested.

- o Line 113: We report that VLP immunization significantly **increased neutralizing antibody titers** (rather than benefits) in both RSV-seropositive pregnant females and their offspring. (Subsequent text describes the other "benefits", but does not specifically mention neutralizing antibody.)

It was corrected as requested.

- Results:

- o Line 143: ...then sacrificed 4 days later **to assess the**

protective efficacy and safety of the VLP vaccines.

o Line 291: These data clearly demonstrate the safety of vaccination with VLPs during pregnancy **in a cotton rat model** and corroborate the strong reduction of lung inflammation in the same vaccinated groups.

It was corrected as requested

• Discussion conclusions:

o Line 301: Furthermore this type of vaccine may not be readily approved for **young** infants.

It was corrected as requested.

o Line 302: Fourth, effects of **naturally** transferred anti-RSV maternal antibodies on **active infant** vaccination efficacy are unknown.

It was corrected as requested.

o Line 319: It may be true that testing of other clinical stage RSV vaccines in RSV-seropositive pregnant animals has not been published, but the authors cannot claim that it has not been done. Modify your statement or delete.

The statement has been deleted.

o Line 387: Overall, our data demonstrate that RSV F/G VLP vaccination of **cotton rats** during pregnancy robustly stimulates pre-existing RSV immunity **with only one immunization, provided** enhanced protection of offspring, **and reduced pulmonary inflammation otherwise associated with RSV infection.**

It was corrected as requested.

Appropriateness of statistics and treatment of uncertainties

• General study design: Please clarify why the mock and RSV i.m. group sizes are smaller than the VLP group sizes. Better power for statistical comparisons could have been achieved if mock and

RSV i.m. group sizes were equal to the rest.

Additional experiments using mock-vaccinated animals were included in the analysis. Animals vaccinated with live RSV i.m. have been analyzed extensively in the same experimental scheme and published in (Blanco et al., Vaccine 2015; Blanco et al., Vaccine 2017). The females and litters included in this work for the unprimed or primed groups vaccinated with live RSV i.m. showed responses that are similar to the previously published studies.

- Please state what statistical software was used for analyses.

The software used are now included in Material and Methods.

- Please consistently refer to either SE or SEM throughout report.

It was corrected as requested.

- Figure 1A would be improved by labeling the week 13, 14, 16 bleeds in the mothers.

It was corrected as requested.

Lines 526-529 / Figure 1B:

- Suggest that the text be revised to clarify that "At each time point, comparison of the expression between..."

It was corrected as requested.

- Based on Supplemental Figure 3, it appears that one of four animals in the primed mock group has particularly high NA titer values. The non-parametric Kruskal-Wallis test is more robust to outliers and may be useful to assess sensitivity of the results to inclusion of this animal.

Analysis using the non-parametric Kruskal-Wallis test has been used.

- Did the authors consider other methods (vs Newman-Keuls) for post hoc testing of pairwise comparisons? For example, the

Tukey method is more robust for unequal group sizes and could provide better Type 1 error control for comparisons among 5 groups. Newman-Keuls may lead to anti-conservative inference in some scenarios (p values smaller than they should be). What is the rationale for using Newman-Keuls post hoc testing? (ref: <https://www.ncbi.nlm.nih.gov/pubmed/22420233>)

ANOVA followed by Tukey *post hoc* testing was used for the statistic analysis of the data presented in this figure to replace the Newman-Keuls *post hoc* analysis of the previous version.

Lines 531-537 / Figure 1C:

- Same comment/question as above for choice of Newman-Keuls vs other methods.

ANOVA followed by Tukey post hoc testing was used for the statistic analysis of the data presented in this figure.

- Suggestion to add error bars to plot.

Error bars were added.

- Why is red color used for markers of RSV i.m.?

The color for the symbols in this graph have been clarified.

- The p value given in the text on line 196 ($p < 0.005$) differs from that shown in the legend for Figure 1C ($p < 0.0005$). Please correct as appropriate.

It was corrected as requested.

Lines 538-547 / Figure 2 / Supplementary Fig 4:

- Please define what Figure 2 error bars represent.

It was corrected as requested.

- Please clarify what statistical methods were used to generate the p values in Supp Fig 4. Were data log-transformed for analysis? They are shown as log₁₀ titers in Figure 2, and ng/ml in Supp Fig 4. Additionally, samples were pooled by group and

assayed multiple times to generate the data presented. The resulting very small group sizes (3-5 assay runs contributing to each mean), warrant an assessment of whether one-way ANOVA assumptions are met. Consider using Kruskal-Wallis test as a non-parametric equivalent to one-way ANOVA as either the primary analysis or as a sensitivity analysis.

It was corrected as requested.

Lines 549-561 / Figure 3

- Please clarify how litter effects were handled in statistical analyses. It appears that the analysis was conducted with the pup as the experimental unit of analysis when in fact it should be the litter. Using the pup as the statistical unit of measure for this study design will generally inflate the Type 1 error rate, leading to anti-conservative inference (p values smaller than they should be). Use of a mixed effects model (to account for correlation among pups in the same litter) would be ideal. Calculating the mean of animals within the same litter, then conducting ANOVA (or similar method) on the litter means would be acceptable as well.

(ref: <https://www.ncbi.nlm.nih.gov/pubmed/23522086>)

Thank you for this suggestion. Litters' means instead of means from all pups born from a group are now analyzed throughout the manuscript.

- Scatter plots with SE bars are preferable for Fig 3A, B, C.

It was corrected as requested.

Lines 563-578 / Figure 4

- Figure 4A. Same issue as above: it is problematic to use the pups rather than the litters as the experimental use of analysis. In addition, pathology score is an ordinal discrete measure, and ANOVA, while robust to departures from normality for large group sizes, may not be so for these small group sizes. Recommend using a non-parametric test instead.

Litters are now used for the analysis and a non-parametric test

was used.

- Figure 4B. Same issue with using pups as the statistical unit of analysis. Additionally, this plot could be improved by using a scatter plot (with mean & SE bars) rather than bar chart. This would allow the reader to evaluate the distribution of the data and appropriateness of statistical methods that assume normality.

Litters are not used for the analysis. Scatter plot was also included.

References

The references cited are appropriate with the following suggestions/requests.

- Consider replacing Nair reference now that the updated RSV global burden data has been published. (Shi et al. Lancet in press, available online July 7, 2017) [http://dx.doi.org/10.1016/S0140-6736\(17\)30938-8](http://dx.doi.org/10.1016/S0140-6736(17)30938-8)

The Nair et al. reference was replaced by the newest Shi et al. reference.

- Line 73: Provide reference(s) that RSV maternal Ab likely protects infants in the first months of life.

References have been added.

- Reference 10 is incomplete. Please update with publication information.

Reference 10 has been completed.

REVIEWERS' COMMENTS:

Reviewer #2 (Remarks to the Author):

The authors have made several revisions which strengthen their report:

1. Addition of pre- and post F protein vaccine comparator groups
2. mAb characterization of RSV F conformation for the VLP and F protein vaccine materials,
3. Additional experiments increased number of data points per vaccination or control group, increasing robustness of analysis,
4. Statistical analysis was modified where appropriate.

Further suggestions to authors:

- Fig 1 C legend has a typo and says "Each symbol represents one litter..." rather than one dam. Correct figure legend.
- Supplementary Fig 5 shows Groups C-J received 100 microgram doses of vaccine. In the methods you clarify groups C-F received 100 micrograms of VLPs, and groups G-J received 10 microgram doses of RSV F. Do you mean to show 100 microliter doses? Correct Supp Fig 5 one way or the other.
- Proofread your report before publication. You have "pre-existenting", "pre-existing" and "preexisting" in the first several pages.

Completing that, it is my opinion your report contributes new and relevant information to the field, and is ready for publication.

Responses to the Reviewers.

I would like to thank the Reviewers for accepting our previous revisions/changes and, to Reviewer #2, now indicating the readiness of the manuscript for publication.

The following are our responses to each of Reviewer #2's suggestions.

Fig 1 C legend has a typo and says "Each symbol represents one litter..." rather than one dam. Correct figure legend.

Figure 1C legend (now Figure 3 legend) has been change as indicated (line 928 of the corrected manuscript).

Supplementary Fig 5 shows Groups C-J received 100 microgram doses of vaccine. In the methods you clarify groups C-F received 100 micrograms of VLPs, and groups G-J received 10 microgram doses of RSV F. Do you mean to show 100 microliter doses? Correct Supp Fig 5 one way or the other.

Supplementary Fig. 5 (now Supplementary Table 1) was updated indicating that 10 micrograms of soluble F proteins were used for vaccination.

Proofread your report before publication. You have "pre-existenting", "pre-existing" and "preexisting" in the first several pages.

The report has been proofread and "pre-existing" now can be found throughout the manuscript.